# Even Pacing Is Associated with Faster Finishing Times in Ultramarathon Distance Trail Running—The “Ultra-Trail du Mont Blanc” 2008–2019

**DOI:** 10.3390/ijerph17197074

**Published:** 2020-09-27

**Authors:** Daniel Suter, Caio Victor Sousa, Lee Hill, Volker Scheer, Pantelis Theo Nikolaidis, Beat Knechtle

**Affiliations:** 1Institute of Primary Care, University of Zurich, 8091 Zurich, Switzerland; dapesu@bluewin.ch; 2Bouve College of Health Sciences, Northeastern University, Boston, MA 02115, USA; cvsousa89@gmail.com; 3Division of Gastroenterology & Nutrition, Department of Pediatrics, McMaster University, Hamilton, ON L8N 3Z5, Canada; hilll14@mcmaster.ca; 4Ultra Sports Science Foundation, 109 Boulevard de l’Europe, 69310 Pierre-Bénite, France; v.scheer@ultrasportsscience.org; 5School of Health and Caring Sciences, University of West Attica, 12243 Athens, Greece; pademil@hotmail.com; 6Exercise Physiology Laboratory, 18450 Nikaia, Greece; 7Medbase St. Gallen, Am Vadianplatz, 9001 St. Gallen, Switzerland

**Keywords:** pacing, trail running, sex difference, ultramarathon

## Abstract

In recent years, there has been an increasing number of investigations analyzing the effects of sex, performance level, and age on pacing in various running disciplines. However, little is known about the impact of those factors on pacing strategies in ultramarathon trail running. This study investigated the effects of age, sex, and performance level on pacing in the UTMB^®^ (Ultra-trail du Mont Blanc) and aimed to verify previous findings obtained in the research on other running disciplines and other ultramarathon races. Data from the UTMB^®^ from 2008 to 2019 for 13,829 race results (12,681 men and 1148 women) were analyzed. A general linear model (two-way analysis of variance (ANOVA)) was applied to identify a sex, age group, and interaction effect in pace average and pace variation. A univariate model (one-way ANOVA) was used to identify a sex effect for age, pace average, and pace variation for the fastest men and women. In our study, pace average and a steadier pace were positively correlated. Even pacing throughout the UTMB^®^ correlated with faster finishing times. The average pace depended significantly on sex and age group. When considering the top five athletes in each age group, sex and age group also had significant effects on pace variation. The fastest women were older than the fastest men, and the fastest men were faster than the fastest women. Women had a higher pace variation than men. In male competitors, younger age may be advantageous for a successful finish of the UTMB^®^. Faster male runners seemed to be younger in ultramarathon trail running with large changes in altitude when compared to other distances and terrains.

## 1. Introduction

Ultra-trail running has previously been defined as any running event that fulfills the following two criteria: The distance must be longer than the classical marathon distance of 42.195 km, and the race must take place in a natural environment in an open country that is mainly off-road [1,2]. In order to further clarify the terms used to describe the distinct off-road running categories, Scheer et al. recently published a consensus statement [3]. In their publication, ultramarathon running is defined by distance as any race longer than the traditional marathon distance with no restrictions regarding the terrain. The authors recommend avoiding the term ultramarathon without precise definition of further details such as surface, elevation change, and level of support as each ultramarathon is unique. Trail running, on the other hand, is defined by the running surface with ≤20–25% of paved or asphalted road [3]. Elevation and race distance are not specified.

In the last years, the various off-road running disciplines have become more popular with an increasing number of athletes participating, and a steady increase in the number of races held each year [4,5]. Additionally, trail and ultramarathon running have become more accessible to non-professional runners [6], despite the high risk of extreme fatigue and/or exhaustion and other potential medical issues that may occur during each race [7]. Among the different ultramarathon races, the Ultra-Trail du Mont Blanc (UTMB^®^), with its 171 km distance, over 10,000 m of vertical gain, and a time limit of 46 h, is one of the most important ultramarathons held in Europe [2]. With six International Trail Running Association (ITRA) points, the UTMB^®^ is in the highest category of trail running worldwide [2,8]. Every year, approximately 2300 qualified runners participate in the race. Accordingly, we consider the UTMB^®^ as highly suitable for the analysis of pacing in trail running over the ultramarathon distance.

Several studies have shown the importance of an athlete’s pacing strategy on performance [9,10,11]. Furthermore, optimal pacing strategies for ultramarathons have been investigated in previous publications [9,10,12]. For example, Hoffman et al. [10] were able to show that an even pacing over the entirety of a 161 km long ultramarathon resulted in the fastest finishing times. This finding concurs with the findings of Knechtle et al. [12] about pacing strategies during a 100 km ultramarathon. Furthermore, a review published by Abbiss et al. [9] suggested an even pacing during a prolonged period of power output to be favorable. In recent years, there has been an increasing number of studies analyzing the effect of sex, performance level, and age on pacing [13,14]. It could be shown that, on average, ultramarathon runners are older than marathon runners. They also have a higher weekly training volume, but train at a lower speed. The experience in ultramarathon events seems to have a great influence on success. Furthermore, a low BMI and a low body fat percentage proved to be advantageous [15]. However, little is known about the effects of sex, age and performance level on pacing strategies specifically in ultramarathon running.

Aging seems to have a significant effect on pacing in endurance sports, and many studies have shown varying pacing strategies for different age groups in marathon and half-marathon runners [16,17,18,19]. Runners with age older than 30 years appear to have a more even distribution of pacing throughout a marathon when compared with younger contestants [17]. These findings, however, have not yet been explored for ultramarathon distance running. Varesco et al. [20] investigated the impact of age on performance in the UTMB^®^ and found that the average speed progressively decreased with age.

Another variable impacting the pacing strategy of runners is their sex. It is well known that, in general, male athletes achieve faster finishing times in running competitions than female athletes [21]. Regarding pacing, women have shown to be better pacers with a more even pace throughout a race [17,22,23]. This effect could also be shown in two studies on ultra-triathlons [24,25] and one on a 100-km marathon [22]. However, to our knowledge, there has not yet been a publication observing the correlation between sex and pacing in an ultramarathon with such great elevation gain as the UTMB^®^.

Furthermore, the performance level of athletes seems to play an essential role in pacing during running competitions. More experienced, faster runners are more constant pacers independent of the running distance, as shown in several studies [16,18]. Competing in the UTMB^®^ involves running on a rougher terrain than in most half-marathons, marathons, or even ultramarathons with over 10,000 m of vertical gain.

Based on existing findings and a lack of knowledge for ultramarathon running on rough, mountainous terrain, the present study aimed to analyze how age, sex, and performance level of athletes affect their pacing during the UTMB^®^, and to identify ideal pacing strategies for ultramarathon distance trail running considering these variables. With rising average participant age and an increasing number of non-professional participants, providing insight into possible pacing strategies for different groups of runners seems to be of particular importance. Considering previous findings [10,12], we hypothesized that a more constant pacing throughout the UTMB^®^ would lead to better results (i.e., final ranking) and, therefore, a higher performance level. The rough terrain and high vertical gain of the UTMB^®^ were potential factors influencing pacing strategies. Taking these distinct conditions into account, we hypothesized the distribution of pacing over the race to vary more when compared to other race categories. Furthermore, we considered a successful pacing strategy to potentially be of even more critical importance for the outcome in a longer, more strenuous race, possibly accentuating previous findings.

## 2. Materials and Methods

### 2.1. Ethical Approval

This study was approved by the Institutional Review Board of Kanton St. Gallen, Switzerland, with a waiver of the requirement for informed consent of the participant as the study involved the analysis of publicly available data (EKSG 01-06-2010). The study was conducted in accordance with the recognized ethical standards according to the Declaration of Helsinki (2013).

### 2.2. Race Description

The Ultra-trail du Mont-Blanc (UTMB^®^) has grown since its first realization in 2003 to be one of the most important ultramarathons in Europe. The race is held annually in France in the region of Chamonix during the last week of August. Participants run along the Tour du Mont Blanc hiking path, a 170 km long circular trail with over 10,000 m of vertical gain, and cross three countries (France, Italy, and Switzerland), seven valleys, 71 glaciers and approximately 400 summits (see race profile of the 2019 edition in Figure 1). While the path usually takes hikers seven to ten days to finish, elite runners, achieve finishing times of ~21 h. The time limit for completion of the race is 46 h and 30 min. All athletes must pass the 15 specific checkpoints within a defined time to be allowed to continue the race. Participants not finishing the entire race or not passing through the checkpoints within the determined times are excluded.

As an ultra-trail held in exposed, mountainous terrain, the UTMB^®^ requires the ability to manage extreme conditions as well as autonomy in the mountains. The race considers the principle of semi-autonomy, defined as autonomy between two aid stations. Aid stations are equipped with food and drink supplies as well as medical facilities in some cases. Participants must at least carry all the items listed on the official UTMB^®^ website [26]. Since 2006, the number of participants is restricted to 2300 runners. To register as a participant, athletes must have scored a minimum of 10 ITRA (International Trail Running Association) points in two years and a maximum of two races. As ultra-trail running has become increasingly popular since the launch of the UTMB^®^ in 2003, there is a rising number of runners fulfilling the inclusion criteria. A lottery system was established in 2007 to prevent overcrowding. In the interest of equitable recognition of top athletes, the race organizers reserve places for elite runners. Requirement for inclusion in this category is an ITRA performance index of over 790 points for males and over 670 points for female runners, respectively.

### 2.3. Data Acquisition

For this study, we collected publicly available data from the official UTMB^®^ website [26]. We included the split times of all participants, independent of age group, sex, and performance level. In addition to the split times, data regarding participants’ name, age, country of origin, finishing time, sex, performance level, and average running speed was gathered from the website of the Deutsche Ultramarathon Vereinigung e.V. (German Ultramarathon Union, DUV) [5]. The database from DUV exclusively consists of finisher data. In contrast, the UTMB^®^ database includes finishers as well as non-finishers but lacks information on the exact age of athletes and their performance level. As the aim of this study was to analyze the impact of age, sex, and performance level on pacing, it was necessary to combine data from both databases and therefore exclude all non-finishers from our evaluation. The data was manually recorded in Microsoft EXCEL sheets for later statistical analysis.

To calculate the pacing of the individual athletes over time, we determined the distance and vertical gain between the different time stations. Time stations seemed to vary between the years due to several different reasons, such as weather conditions, and only the route data for the year 2019 was publicly available at the moment of acquisition. Thus, we were forced to exclude sometime stations and even data from whole years from the final analysis. This excluded data was comprised of all-time stations that we could not compare directly and all years, which showed less than 18 time stations directly comparable with the year 2019 (cf. Table 1). The cut-off of a minimum of 18 comparable time stations was defined based on a detailed evaluation of the available data. We reached for the optimal trade-off between included years and time stations. Table 1 shows the amount of includable years according to different cut-off values for the minimum number of time stations. Including all available time stations would have resulted in an analysis of merely a single year. In contrast, the inclusion of all years would have reduced the amount of comparable time stations to two. Since our study focused on pacing, we decided that a higher quantity of time stations and hence an improved observed ability of variation in pacing would be of great value for our statistical analysis. At the same time, we aimed to include as many years as possible without reducing the validity of our results. In synopsis of these considerations, we decided on the cut-off of 18 comparable time stations, hence including nine years in our final analysis.

### 2.4. Statistical Analysis

Data were tested for normality with the K-S test, and after showing parametric (*p* > 0.05) distribution data were then expressed as mean and standard deviations. Pace variation was determined as the coefficient of variation (CV%) of pace average between different time station throughout the race. CV% was calculated individually for each athlete based on the pace throughout the race. A general linear model (two-way ANOVA) was applied to identify a sex, age group, and interaction effect in pace average and pace variation. A univariate model (one-way ANOVA) was applied to identify a sex effect for age, pace average, and pace variation for the fastest men and women. Pearson’s correlation coefficient was applied. Correlation coefficient descriptors were considered as small: 0.1–0.3; moderate: 0.3–0.5; and large: >0.5 [27]. Statistical significance was set *p* < 0.05. Statistical Software for the Social Sciences (IBM SPSS Statistics version 26.0 for Windows, IBM Corp, Armonk, NY, USA, 2018) and GraphPad Prism (GraphPad Prism version 8.4.0 for Windows, GraphPad Software, La Jolla, CA, USA, 2018) were used to carry out the analysis.

## 3. Results

A total of 13,829 race results were analyzed (n = 13,829), achieved by 12,681 men and 1148 women. The average pace depended significantly on sex and age group for time station 18 (TS18) (F = 10.2, *p* = 0.007, _p_η^2^ = 0.445; F = 5.48; *p* = 0.013, _p_η^2^ = 0.860, respectively), TS20 (F = 13.6, *p* < 0.001, _p_η^2^ = 0.099; F = 10.0; *p* < 0.001, _p_η^2^ = 0.910), TS22 (F = 7.4, *p* = 0.006, _p_η^2^ = 0.328; F = 17.5; *p* < 0.001, _p_η^2^ = 0.949), and TS24 (F = 4.1, *p* = 0.044, _p_η^2^ = 0.101; F = 6.8; *p* < 0.001, _p_η^2^ = 0.901), but no interaction between sex and age group was identified. Similar results were obtained when only considering the top five in each age group: TS18 (F = 40.7, *p* < 0.001, _p_η^2^ = 0.794; F = 9.3; *p* = 0.004, _p_η^2^ = 0.915), TS20 (F = 80.5, *p* < 0.001, _p_η^2^ = 0.923; F = 28.3; *p* < 0.001, _p_η^2^ = 0.971), TS22 (F = 104.5, *p* < 0.001, _p_η^2^ = 0.923; F = 60.0; *p* < 0.001, _p_η^2^ = 0.990), and TS24 (F = 36.4, *p* < 0.001, _p_η^2^ = 0.823; F = 16.5; *p* < 0.001, _p_η^2^ = 0.994); with significant sex × age group interactions in TS18 (F = 3.0, *p* = 0.005, _p_η^2^ = 0.091) (Figure 2). Performance seems to decrease linearly for men, regardless of TS, whereas for women, there is an apparent optimal age between 30 and 40 years old for TS 20. When considering only top five athletes, both men and women seem to have an optimal age around 30 to 40 years old, regardless of TS.

Age group had a significant effect on pace variation only in TS20 (F = 5.3, *p* = 0.015, _p_η^2^ = 0.842) when including all athletes in the analysis. When considering the top five in each age group, pace variation depended significantly on sex and age group in TS18 (F = 6.0, *p* = 0.015, _p_η^2^ = 0.230; F = 2.0; *p* = 0.049, _p_η^2^ = 0.533), TS20 (F = 14.9, *p* < 0.001, _p_η^2^ = 0.521; F = 6.0; *p* < 0.001, _p_η^2^ = 0.779), and TS24 (F = 22.5, *p* < 0.001, _p_η^2^ = 0.628; F = 4.9; *p* < 0.001, _p_η^2^ = 0.746); significant sex ×age group interactions were observed in TS20 (F = 2.2, *p* = 0.047, _p_η^2^ = 0.063) (Figure 3). There is no apparent trend in pace variation across age groups for men and women in TS18, TS20, and TS22, whereas in TS24 both sexes around 30 to 40 years old seem to have a higher pace variation. When considering only top five athletes, male and female athletes aged 30 to 40 have the lower pace variation across age-groups. 

The comparison between men and women with only top five finishers in each race showed that the fastest women were older than the fastest men in TS20 (F = 7.1, *p* = 0.013) and TS22 (F = 8.8, *p* =0 .008). The fastest men were faster than the fastest women in TS18 (F = 38.5, *p* < 0.001), TS20 (F = 76.2, *p* < 0.001), TS22 (F = 33.3, *p* < 0.001), and TS24 (F = 61.8, *p* < 0.001). Women had a higher pace variation than men in TS18 (F = 7.0, *p* = 0.013), TS20 (F = 5.2, *p* = 0.030), and TS24 (F = 7.0, *p* = 0.029) (Figure 4). Pace average and a steadier pace were positively correlated for men, women, and all together. Although age was also significantly correlated with pace average for men, women, and overall, for men and overall analysis, the correlation was weak (Table 2).

## 4. Discussion

This study investigated the effect of age, sex, and performance level on pacing specifically for the UTMB^®^, with the hypothesis of even pacing to be the most successful strategy. Furthermore, we aimed to verify previous findings obtained in the analysis of other running categories and other ultramarathons. There were two main findings of this study. Firstly, little pacing variation, and hence a more even pace throughout the race, was associated with a faster average pace, regardless of the sex of the athlete. This finding complies with our hypothesis that an even pace is favorable during long-distance runs. A steady pace in marathons or ultramarathons is considered the best strategy for success [9,12]. The UTMB^®^ includes more elevation gain when compared to many other ultramarathons or marathons, suggesting that maintaining a constant speed might prove to be a greater challenge. One reason for the observed small correlation between even pacing and faster average pace could be that elevation gain and loss potentially cancel each other out throughout the race. Secondly, we found that the fastest men in the UTMB^®^ were younger than the fastest men as reported for other ultramarathons [28,29] and that the fastest men competing in the UTMB^®^ were younger than the fastest women participating in the race.

### 4.1. Age and Pacing

When investigating the effects of age on pacing, we found different age groups to be faster, depending on their sex. The fastest age group for men was 30–34 years, whereas, for women, there were two peaks with ages 35–39 and 40–44 years. Overall, the fastest men were, therefore, younger than the fastest women. This is an interesting finding, as other studies have shown the fastest ultramarathon competitors to be 35 years old on average, regardless of their sex [28,29]. Our observation that the fastest men were younger than the fastest women could be attributed to age-related declining testosterone [30,31] and the associated decrease in maximum oxygen capacity and relative muscle mass [32,33,34]. Especially in runs with a lot of altitude difference like the UTMB^®^, the cumulative strain on the osteoarticular and musculotendinous systems is high. Gajda et al. [35] showed in their study that the stress on joints and musculature plays a large role when competing in long distance runs. Therefore, strategies that minimize the forces on the mentioned structures are certainly one of the decisive factors for faster race times. Millet et al. [36] showed in their publication that fractional use of VO_2_max at the expense of economy can be worthwhile, if it reduces the load on the musculotendinous and osteoarticular system. Since VO_2_max decreases with age and considering these findings, younger men would be at an advantage because the previously mentioned mechanism allows them to reduce the load on joints and muscles over the entirety of the race distance. The importance of economy in long distance runs is shown by the study of Berger et al. [37], in which the authors investigated the physiological adaptation during a successful world record run in the ultra-distance range. In addition, Sloniger et al. [38,39] showed in their studies that running uphill required more muscle work, and thus, a higher relative muscle mass compared to running on flat ground. At the same time, a higher maximum oxygen capacity proved to be advantageous [38,39]. When the testosterone level drops with age, muscle mass as well as maximum oxygen capacity decrease [30], resulting in an advantage for younger uphill runners over their older counterparts. It is important to note that a high muscle mass tends to negatively affect performance in an ultramarathon simply because of the proportionally high cost of running uphill. We therefore estimate the contribution of VO_2_max to our observation as the main factor leading to the benefit of young age in males competing in the UTMB^®^. In women, no hormones of equal relevance for physical endurance performance decrease with age, so that older competitors are not at the same disadvantage when compared to younger participants. Our finding that younger women were not significantly faster than competitors from a higher age group hence supports our rationale. Our study showed as well that after a certain age, performance and older age were negatively correlated. This age and performance correlation is well known and is possibly due to various changes in physiological parameters such as maximum oxygen consumption (VO_2_max), maximal heart rate, stroke volume, lactate threshold, economy of movement, muscle fiber type, activity of aerobic enzymes, blood volume, and skeletal muscle mass [40] associated with endurance performance capacity that typically occur with aging [28,41].

### 4.2. Sex and Pacing

As previously observed in other studies, men were faster than women regardless of age group or performance level in our analysis. This is most probably due to the differences in male and female physique [42,43,44]. An interesting finding was that in contrary to our hypothesis and previous results [23,45], women competing in the UTMB^®^ showed a higher pace variation than men. This could be because more men participated in the race (12,681 men and 1148 women). Another possible reason could be that in terrain with more variation of altitude, men are potentially able to run uphill faster due to their higher relative muscle mass compared to fat mass [34,42,43]. The small number of women participating in the race, making up merely 11% of all finisher results, is striking. Women may experience additional social pressures not affecting men, such as negotiating traditional gender roles and motherhood. This could potentially contribute to variations in performance [46]. However, the almost tenfold difference in participation between the sexes could also be a result of the ITRA points system limiting participation.

### 4.3. Performance Level and Pacing

In our analysis, no significant correlation between performance level and pacing was observed. A possible reason could be the already high level of performance level needed to be able to compete in the UTMB^®^. Since a high-performance level is already required for registration, we assume that the differences in performance levels of the individual athletes were minor, and therefore, no correlation was observed. The previously mentioned exclusion of participants not passing the checkpoints or finishing the race in the defined times contributes to this selection of athletes. It should also be mentioned that the UTMB^®^, unlike many other races, has a great variability in terms of terrain and environmental conditions within the race. Not all competitions that athletes use to train for and gain access to the UTMB^®^ prepare them equally well. Since the performance level is measured by the number of races completed, athletes with the same performance level may have different levels of preparation for the UTMB^®^. A possible correlation between pacing and performance level might thus be concealed. Further studies, including ultramarathons that display a greater variation in the performance level of the athletes, would be necessary to find a potential effect of performance level on pacing strategy. However, a valid analysis could prove to be difficult, considering the minimum performance levels required for all races in this category. Furthermore, the choice of races completed in the course of the UTMB^®^ qualification might be of great importance for the success in the race.

### 4.4. Strengths, Weaknesses, Limitations, Implications for Future Research, and Practical Applications

The main limitation of this study proved to be the different courses of the race throughout the different years, which hindered a direct comparison of all years and can also be made responsible for the main data loss. We needed to rule out otherwise usable data. Despite the cuts, we had an extensive data set to our disposal for the statistical analysis. Although the UTMB^®^ is one of the most important ultramarathons, it is noteworthy that our findings may not be directly convertible to other races in this category, since the inter-race differences in terrain and elevation are substantial. Additionally, we could not perform more robust association analyses due to the nature of the data, with different time stations every year. Studies investigating other races would hence be necessary to verify our results conclusively. Furthermore, research including less selective ultramarathon running competitions could potentially yield further findings regarding the effects of performance level on pacing.

To our knowledge, this is the first study on pacing strategy exclusively for trail running over the ultramarathon distance. Our main results indicate that runners and trainers should focus on maintaining a steady pacing, even on rough terrain. Furthermore, we found that the classical distribution of age in successful marathon running may not apply to mountainous ultramarathons, especially for men. This implicates that different types of adjuvant training, focusing more on maintaining a high VO_2_max throughout the race and on a high amount of relative muscle mass compared to body fat, might lead to better results in the UTMB^®^.

## 5. Conclusions

To conclude, our study supports the hypothesis that constant (even) pacing is the best strategy to achieve a top position in an ultramarathon running event such as the UTMB^®^, regardless of age or sex. Athletes and coaches should, therefore, focus on a constant distribution of work throughout the race. These findings coincide with conclusions from previous publications [9,12]. In addition, and in contrast to previously published results, we found that for men, younger competitors may have advantages over their older counterparts. This could be due to their physical advantages in terms of altitude in races comparable to the UTMB^®^. More research including a variety of trail runs over the ultramarathon distance would be necessary to underpin this finding.

## Figures and Tables

**Figure 1 ijerph-17-07074-f001:**
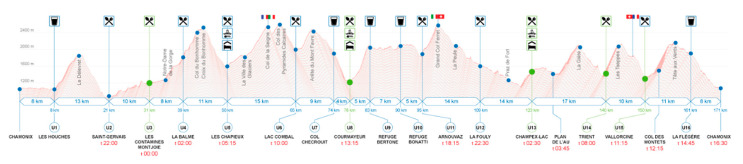
Race profile of the 2019 edition (Ultra-trail du Mont Blanc (UTMB^®^) Website (20)).

**Figure 2 ijerph-17-07074-f002:**
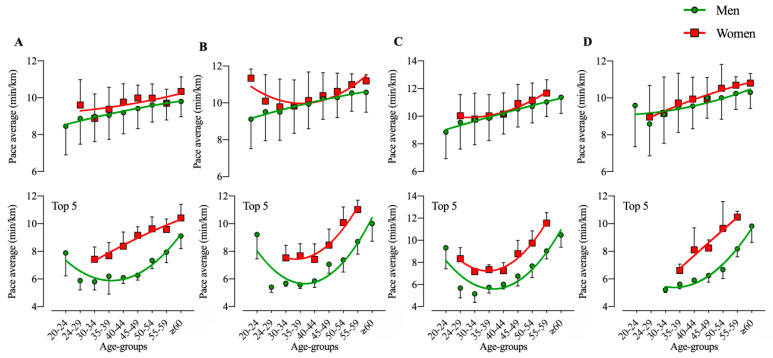
Pace average in different age-groups for all men and women and for top five in each age-group; (**A**): TS18; (**B**): TS20; (**C**): TS22; (**D**): TS24.

**Figure 3 ijerph-17-07074-f003:**
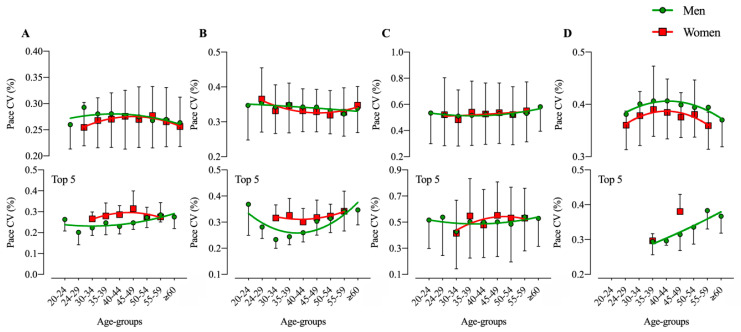
Pace variation in different age-groups for all men and women and for top five in each age-group; (**A**): TS18; (**B**): TS20; (**C**): TS22; (**D**): TS24.

**Figure 4 ijerph-17-07074-f004:**
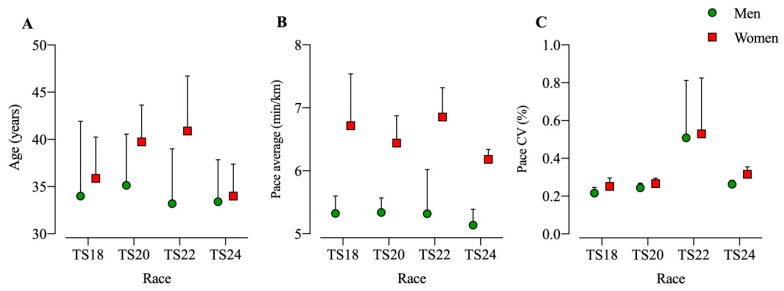
Age, pace average and pace variation for the top five men and women in each race; (A): Age; (**B**): Pace average; (**C**): Pace.

**Table 1 ijerph-17-07074-t001:** Included time stations and years in the present analysis.

# of Included Time Stations	# of Includable Years	% Included Years
24	1	6.7
23	1	6.7
22	3	20.0
21	3	20.0
20	6	40.0
19	6	40.0
18	9	60.0
17	9	60.0
16	10	66.7
15	11	73.3
14	11	73.3
13	12	80.0
12	13	86.7
11	13	86.7
10	13	86.7
9	13	86.7
8	13	86.7
7	13	86.7
6	13	86.7
5	13	86.7
4	14	93.3
3	14	93.3
2	15	100.0

**Table 2 ijerph-17-07074-t002:** Correlation matrix. Data presented as Pearson’s correlations coefficient, r and (*p*-value).

Sex	Pace Variable	Age	Pace Average
Women	Pace average	0.242 (<0.001)	
	Pace variation	–0.026 (0.384)	0.253 (<0.001)
Men	Pace average	0.072 (<0.001)	
	Pace variation	0.002 (0.841)	0.304 (<0.001)
Women and men	Pace average	0.073 (<0.001)	
	Pace variation	0.001 (0.903)	0.299 (<0.001)

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
