# Peer review of "Even Pacing Is Associated with Faster Finishing Times in Ultramarathon Distance Trail Running—The “Ultra-Trail du Mont Blanc” 2008–2019"

_ijerph, 2020, doi:10.3390/ijerph17197074_

Round 1

Reviewer 1 Report

  • Introduction, this section is well written and clear for understanding the research framework and previous studies. I only suggest to improve the plausible reasons of better performances for experiences runners (number of races per year, years running at the highest level, etc. L79-87), the different performances by sex (e.g. anthropometric, biological, physiological aspects, L72-78), and the importance of altitude for long distance races.
  • Statistical analyses, as each athlete is itself a random factor, I encourage the authors to run another model (mixed linear model for repeated measures) with fixed (sex, experience level) and random factors (id athlete). If not the GEE would also fit the data characteristics. In addition, the autocorrelation function (temporal series of stations of each year and the series of years) needs to be included for each sex and cross-correlated with experience level, then the authors would obtain a more reliable and valid results of performance variability (stable or unstable performance) during each year (intra-year) and across the years (inter-years).
  • The ANOVA results need to include the partial eta square (effect size) results.
  • The Pearson results do not add up to the analyses, please include the suggested improvements.
  • Please add the (CV%) after coefficient of variation (L164).
  • Discussion, please after redoing the ACF and CCF results add more arguments to discuss the variability of performance.

Author Response

Introduction, this section is well written and clear for understanding the research framework and previous studies. I only suggest to improve the plausible reasons of better performances for experiences runners (number of races per year, years running at the highest level, etc. L79-87), the different performances by sex (e.g. anthropometric, biological, physiological aspects, L72-78), and the importance of altitude for long distance races.

Answer: We agree with the expert reviewer and had already adjusted the introduction according to the detailed adjustment requests of another expert reviewer, who requested similar adjustments.

Statistical analyses, as each athlete is itself a random factor, I encourage the authors to run another model (mixed linear model for repeated measures) with fixed (sex, experience level) and random factors (id athlete). If not the GEE would also fit the data characteristics. In addition, the autocorrelation function (temporal series of stations of each year and the series of years) needs to be included for each sex and cross-correlated with experience level, then the authors would obtain a more reliable and valid results of performance variability (stable or unstable performance) during each year (intra-year) and across the years (inter-years).

Answer: We agree with the expert reviewer. Unfortunately, we do not have “athlete experience” as an available variable. Time stations also change every year. Thus, any sort of comparison between years or stations would be biased. Additionally, we performed the autocorrelation method, as you suggested, but we could not find reliable results in the dispersion graphs. This is possibly because the Calendar Year is not a linear variable, since there are some years without data.

The ANOVA results need to include the partial eta square (effect size) results.

Answer: We agree with the expert reviewer and added the partial eta square, as suggested.

The Pearson results do not add up to the analyses, please include the suggested improvements.

Answer: We agree with the expert reviewer. We added the further information requested and performed the analysis. Unfortunately, we could not include autocorrelation results in the manuscript due to the aforementioned reasons and included this as a limitation of our results.

Please add the (CV%) after coefficient of variation (L164).

Answer: Added, as requested.

Discussion, please after redoing the ACF and CCF results add more arguments to discuss the variability of performance.

Answer: Please see previous comments.

Reviewer 2 Report

Thank you for the opportunity to review this well prepared paper, I particularly enjoyed the figures, and it's great to see such a high profile race as UTMB, assessed rigorously. Please find below my line by line recommendations: 21 - please consider removing 'specifically' as this currently reads as if these factors have not been investigated at all, yet in the next sentence you suggest that this work builds upon previous works. 33 - 'Even pacing throughout the UTMB® correlated with faster finishing times.' this may work better on line 29 34/37 - I'm not entirely sure what this last sentence means. It seems to mix pace, performance, athlete sex, and environmental factors. 49/50 - this seems like a really clear definition for trail running, is there a supporting reference for it? Is it supported by [3]? If no reference, I agree that this definition seems appropriate, but the language around the definition needs to reflect its novelty. 53 - you've suggested these trends are recent, but have used a reference that's 10 years old. I agree the trends are likely still true, but a more recent reference may support these trends, or a reword may be required 56 - am I right in thinking the distance of 171km has only been used recently? Records appear to show that races have been contested over 88 - 170km? 59 - please amend 2'300 to 2,300 or 2300 66 - you may want to consider amending coincides with concurs here 71 - is there a way to expand this point slightly to better link to the next sentence? Are ultramarathon competitors typically older than those of standard marathons for example? Are there metabolic adaptations that may better align older competitors to ultra? 80 - by perform better here do you mean that male athletes are faster than females? If so, it may be worth being specific as 'perform better' when talking about pacing may refer to even pacing or time from self-selected target time 82 - has this been investigated in other ultra races? If so, please expand and reference accordingly. Again, I think this research is warranted given UTMB's prestige, but the rationale for the paper needs to be more precise than 'x hasn't been done for UTMB) 85 - it may fit better to start the new paragraph here. 163 - please expand your statement regarding normality. Did data pass the test? Were any other tests of normality performed? 164 - please revise the statement regarding coefficient of variation for clarity 169 - please consider rephrasing the statement regarding statistical significance, as it may be construed as unclear what the use of a percentage refers to in this context to some readers. I would really like to see confidence intervals included to capture the uncertainty of the estimate in these data, especially given the weak correlations presented and the wide range of age groups examined. 174 - please double check as ex may supposed to be sex? Also abbreviations are commonly stated in full in text first, then the abbreviation placed in brackets, not vice versa as per this section. 175 - are these results for sex then age, respectively? Figure 2 - interesting to note that these data typically follow a J-shaped curve despite being normally distributed? I hope to see this commented upon in the discussion. Figure 3D - are there only two data points for women here? Discussion comments: Do you feel your first finding is adequately supported given the magnitude of correlations you've presented in the correlation matrix above? I agree there is support for these comments in the figures to some extent, but whilst highly significant, the correlations range from trivial to small, possibly moderate for these effects. The discussion regarding muscle mass and VO2max is well referenced, but fails to acknowledge the potential role for large eccentric components in trail ultra-running, and that cost of running not maximal oxygen uptake is likely a more important factor given the prolonged duration of UTMB and similar races. The work of Guillaume Millet and Kristine Snyder provide useful starting points for discussions of mechanical work and gradients, and the role of the cost of running in ultra-marathon performance is also discussed by Guillaume and Gregoire Millet quite convincingly e.g. https://journals.physiology.org/doi/full/10.1152/japplphysiol.00642.2012 243/244 - as above, you have stated here that the correlation between age and declines in performance are well known, but you have said that you found your findings to be surprising and that they run somewhat counter to the literature. Please rephrase. Again, it must be emphasised that the correlation in the present work is small-moderate, I am not discounting the importance of a small correlation in such a large cohort, but feel this strengthens the argument for the inclusion of confidence intervals around your correlations. I would also recommend finding a potentially stronger reference than the current 34 to support these claims, as whilst it is interesting is a focussed review on stem cell activity in geriatric populations, and so does not examine the same ageing process as outlined in the present paper, nor that is likely to be experienced by aged UTMB participants 257 - is higher muscle mass an advantage when running uphill, or is higher relative muscle mass compared to fat mass the advantage between sexes? 260 - in the present dataset is this likely an explanation of performance or participation? Given the nearly 10 fold difference between sexes in participant numbers it may suggest the latter, and this may be a product of the ballot and scoring system that permits entry too? This would transition well into your next subsection 278 - can you be more specific here? The UTMB also shows a great deal of intra-race variability, which may not be present in other races that employ a fixed route over a more consistent surface or environmental conditions. This is important to acknowledge, as the races athletes are using to qualify for UTMB may not best prepare them for success in UTMB, thus athlete race selection, training and other preparatory measures need to be considered based off your work. 283 - here you talk about reliability, and use of your findings but in the previous section you've just mentioned how the data themselves may be unreliable or lack transferability for various reasons. This is a consistent theme throughout the manuscript, and again I think it comes back to the magnitude of the correlations seen. Ultimately, I think what these correlations point to is that while young male athletes may be fastest for a variety of reasons, simply being properly prepared for UTMB leads to completion and faster performance, regardless of sex or age...especially given you have only analysed race finishers. 288 - again, I'm not convinced that muscle mass per se is the key factor here, especially given the terrain, altitude and technicality of a race like UTMB. A lower body fat percentage and therefore a higher muscle mass percentage may increase VO2max, but if muscle mass is too large this will increase the cost of running and the mechanical cost associated with moving said mass uphill and potentially incur a greater cost during downhill running too, as more mass needs to be dissipated over the same surface area (i.e. heavier weight on same size feet). On the whole, the paper is well written, and the analyses conducted are impressive, but I strongly feel that the discussion does not fully elucidate the potential explanations for and the magnitude of the results obtained. The tone needs to be diluted slightly, as the reader could be mistaken for thinking the authors had found much stronger correlations. At present the paper feels a little incomplete and a more inclusive discussion is warranted.

Author Response

Thank you for the opportunity to review this well prepared paper, I particularly enjoyed the figures, and it's great to see such a high profile race as UTMB, assessed rigorously. Please find below my line by line recommendations:

21 - please consider removing 'specifically' as this currently reads as if these factors have not been investigated at all, yet in the next sentence you suggest that this work builds upon previous works.

Answer: We agree with the expert reviewer and have cancelled the word “specifically” from the sentence.

33 - 'Even pacing throughout the UTMB® correlated with faster finishing times.' this may work better on line 29

Answer: We agree with the expert reviewer and have adjusted the sentence order accordingly.

34/37 - I'm not entirely sure what this last sentence means. It seems to mix pace, performance, athlete sex, and environmental factors.

Answer: We agree with the expert reviewer and have changed the sentence accordingly. This passage was meant to further emphasize the finding.

49/50 - this seems like a really clear definition for trail running, is there a supporting reference for it? Is it supported by [3]? If no reference, I agree that this definition seems appropriate, but the language around the definition needs to reflect its novelty.

Answer: We agree with the expert reviewer and have found the definition to be supported by [3]. We have made the adjustment.

53 - you've suggested these trends are recent, but have used a reference that's 10 years old. I agree the trends are likely still true, but a more recent reference may support these trends, or a reword may be required 56 - am I right in thinking the distance of 171km has only been used recently? Records appear to show that races have been contested over 88 - 170km?

Answer: We agree with the expert reviewer and have used a more recent reference. Yes, the course of the UTMB was altered from year to year due to environmental factors. Only in the last few editions, the distance was 171km.

59 - please amend 2'300 to 2,300 or 2300

Answer: We agree with the expert reviewer and have replaced “2’300” by “ 2300”.

66 - you may want to consider amending coincides with concurs here 71 - is there a way to expand this point slightly to better link to the next sentence? Are ultramarathon competitors typically older than those of standard marathons for example? Are there metabolic adaptations that may better align older competitors to ultra?

Answer: We agree with the expert reviewer and have extended the point with the description of an ultramarathon runner according to the description published in the article “Physiology and Pathophysiology in Ultra-Marathon Running”.

80 - by perform better here do you mean that male athletes are faster than females? If so, it may be worth being specific as 'perform better' when talking about pacing may refer to even pacing or time from self-selected target time

Answer: We disagree with the expert reviewer. Our point is that, in general, male athletes have an advantage in sports and that overall results are better, independent of sports category (although, of course, differences are greater in some sports).

82 - has this been investigated in other ultra races? If so, please expand and reference accordingly. Again, I think this research is warranted given UTMB's prestige, but the rationale for the paper needs to be more precise than 'x hasn't been done for UTMB)

Answer: We agree with the expert reviewer and have extended this point, integrating two additional papers about ultra- triathlons and one publication analyzing a 100km ultramarathon.

85 - it may fit better to start the new paragraph here.

Answer: We agree with the expert reviewer and have adapted the layout accordingly.

163 - please expand your statement regarding normality. Did data pass the test? Were any other tests of normality performed?

Answer: The test applied is the adequate for our sample size. More information regarding the results of the normality test was added, as requested.

164 - please revise the statement regarding coefficient of variation for clarity

Answer: The sentence was rephrased and expanded for clarity, as requested.

169 - please consider rephrasing the statement regarding statistical significance, as it may be construed as unclear what the use of a percentage refers to in this context to some readers. I would really like to see confidence intervals included to capture the uncertainty of the estimate in these data, especially given the weak correlations presented and the wide range of age groups examined.

Answer: We agree with the expert reviewer and rephrased the sentence.

174 - please double check as ex may supposed to be sex? Also abbreviations are commonly stated in full in text first, then the abbreviation placed in brackets, not vice versa as per this section.

Answer: We agree with the expert reviewer and have replaced “ex” by “sex”. The abbreviations were also changed as requested.

175 - are these results for sex then age, respectively? Figure 2 - interesting to note that these data typically follow a J-shaped curve despite being normally distributed? I hope to see this commented upon in the discussion. Figure 3D - are there only two data points for women here? Discussion comments: Do you feel your first finding is adequately supported given the magnitude of correlations you've presented in the correlation matrix above? I agree there is support for these comments in the figures to some extent, but whilst highly significant, the correlations range from trivial to small, possibly moderate for these effects.

Answer: The results are for sex and age, respectively. The sentence was rephrased for clarity. The J-shaped curve is because the data is being plotted versus age-groups (x-axis) and not distribution. Although we can discuss the nature of the data, this has no impact on our goals and results. Figure 3D: yes. Regarding the correlations, we agree with the expert reviewer and rephrased our interpretation of the results.

The discussion regarding muscle mass and VO2max is well referenced, but fails to acknowledge the potential role for large eccentric components in trail ultra-running, and that cost of running not maximal oxygen uptake is likely a more important factor given the prolonged duration of UTMB and similar races. The work of Guillaume Millet and Kristine Snyder provide useful starting points for discussions of mechanical work and gradients, and the role of the cost of running in ultra-marathon performance is also discussed by Guillaume and Gregoire Millet quite convincingly e.g. https://journals.physiology.org/doi/full/10.1152/japplphysiol.00642.2012 243/244 - as above, you have stated here that the correlation between age and declines in performance are well known, but you have said that you found your findings to be surprising and that they run somewhat counter to the literature. Please rephrase.

Answer: We agree with the expert reviewer and have included a more detailed discussion, highlighting the importance of VO2max as published by G.Y. Millet.

Again, it must be emphasised that the correlation in the present work is small-moderate, I am not discounting the importance of a small correlation in such a large cohort, but feel this strengthens the argument for the inclusion of confidence intervals around your correlations. I would also recommend finding a potentially stronger reference than the current 34 to support these claims, as whilst it is interesting is a focussed review on stem cell activity in geriatric populations, and so does not examine the same ageing process as outlined in the present paper, nor that is likely to be experienced by aged UTMB participants

Answer: We agree with the expert reviewer and have rephrased where needed.

257 - is higher muscle mass an advantage when running uphill, or is higher relative muscle mass compared to fat mass the advantage between sexes?

Answer: We agree with the expert reviewer and have changed the sentence.

260 - in the present dataset is this likely an explanation of performance or participation? Given the nearly 10 fold difference between sexes in participant numbers it may suggest the latter, and this may be a product of the ballot and scoring system that permits entry too? This would transition well into your next subsection

Answer: We agree with the expert reviewer and have discussed the possibility that the ITRA points system for race registration might increase the gender gap in participation.

278 - can you be more specific here? The UTMB also shows a great deal of intra-race variability, which may not be present in other races that employ a fixed route over a more consistent surface or environmental conditions. This is important to acknowledge, as the races athletes are using to qualify for UTMB may not best prepare them for success in UTMB, thus athlete race selection, training and other preparatory measures need to be considered based off your work.

Answer: We agree with the expert reviewer and have discussed the fact, that even though on paper same ITRA-points translate to same performance-level, a correlation might be hidden because of this. The variety of races used to gain the required ITRA points might have a great impact on true performance in the UTMB.

283 - here you talk about reliability, and use of your findings but in the previous section you've just mentioned how the data themselves may be unreliable or lack transferability for various reasons. This is a consistent theme throughout the manuscript, and again I think it comes back to the magnitude of the correlations seen. Ultimately, I think what these correlations point to is that while young male athletes may be fastest for a variety of reasons, simply being properly prepared for UTMB leads to completion and faster performance, regardless of sex or age...especially given you have only analysed race finishers.

Answer: We agree with the expert reviewer and removed the reliability statement due to the low magnitude of the observed correlations. However, in part we do not agree with the expert reviewer. If younger men were faster, it cannot be only because they are better prepared for the UTMB. Older male competitors would have the advantage of experience, because they have potentially more UTMBs behind them.

288 - again, I'm not convinced that muscle mass per se is the key factor here, especially given the terrain, altitude and technicality of a race like UTMB. A lower body fat percentage and therefore a higher muscle mass percentage may increase VO2max, but if muscle mass is too large this will increase the cost of running and the mechanical cost associated with moving said mass uphill and potentially incur a greater cost during downhill running too, as more mass needs to be dissipated over the same surface area (i.e. heavier weight on same size feet). On the whole, the paper is well written, and the analyses conducted are impressive, but I strongly feel that the discussion does not fully elucidate the potential explanations for and the magnitude of the results obtained. The tone needs to be diluted slightly, as the reader could be mistaken for thinking the authors had found much stronger correlations. At present the paper feels a little incomplete and a more inclusive discussion is warranted. 

Answer: We agree with the expert reviewer and have pointed out the importance of the VO2max rather than the overall muscle mass. Also, we have changed “muscle mass” to “relative muscle mass” and adapted the recommendations accordingly.

Round 2

Reviewer 1 Report

Thanks for the revised manuscript 

Author Response

no further changes are required

Reviewer 2 Report

Thank you for providing a revised version of this paper. I have some amendments that I would prefer you to include before recommending the paper be accepted for publication.

Line 82 - as per my previous review, please consider amending the word better here. My understanding is that you mean faster. It is appropriate to be direct here, especially given that in the next sentence you say that women are better at pacing than male athletes. Thank you for the clarification here though, this is nicely written and highlights the aim of the paper clearly.

Statistical analyses - I feel that descriptors of correlation coefficients must be included in this section, I wish I had mentioned and pushed for this in my previous review, but was perhaps not firm enough. My rationale for including descriptors in this section is that it makes Table 2 more clear, and highlights that despite the statistical significance of the findings, the relationships between assessed variables are small/weak at best.

Appropriate references for descriptions of correlations can be found:

Cohen, 1988 - 0.1 - 0.3 Small, 0.3 - 0.5 Moderate and >0.5 large

Hinkle et al., 2003 - 0 - 0.3 negligible, 0.3-0.5 Low, 0.5 -0.7 Moderate, 0.7 -0.9 High, >0.9 Very High

Hopkins et al., 2009 - 0 - 0.1 trivial, 0.1 - 0.3 Small, 0.3 - 0.5 Moderate 0.5 - 0.7 Large 0.7 - 0.9 Very Large, >0.9 Nearly Perfect

My other question regarding this section, is can you analyse top 5 finishers parametrically? By definition, even in a normally distributed data set, the top 5 are extreme outliers, and themselves would not follow a normal distribution, or at least represent an inadequate sample size to make such an assessment. I agree with including them in the paper, and the discussion around these highly placed finishers is worthwhile, but may be statistically flawed.

Thank you for your comments regarding Figure 3D, and further expanding the results section with the inclusion of effect sizes are per partial eta squared. I disagree with the authors' comments that describing the nature of the curves in the figures is of little value, and that is does not agree with the paper's aims. I quote the following from the introduction

'...the present study aimed to analyze how age, sex, and performance level of athletes affect their pacing during the UTMB®, and to identify ideal pacing strategies for ultramarathon distance trail running considering these variables.' 

Describing the curves, as presented by age group (and sex), provides a fuller picture of the data and supports the aim of the paper. If you can see that pace declines more rapidly beyond a certain age, or pace is more variable beyond a certain age, this has the potential to directly influence training and therefore race outcome. Especially, as the authors point out with an increasing participant finishing age, greater participation numbers and the challenging terrain.

Discussion - I feel you have mischaracterised the findings and discussion points of reference 36 here. What Millet and colleagues suggest in this paper, is that strategies that may be considered to increase the fractional utilisation of VO2max, at the expense of running economy, may be worth employing IF they reduce the load experienced by the musculotendinous and osteoarticular systems. Again, this is especially important given the aim and novelty of the current paper, as higher loads are placed upon these tissues, with increasing elevation changes throughout a race. Please amend accordingly. 

Indeed further evidence for economy in prolonged running can be found in a paper recently published in this journal:

Berger, N.; Cooley, D.; Graham, M.; Harrison, C.; Best, R. Physiological Responses and Nutritional Intake during a 7-Day Treadmill Running World Record. Int. J. Environ. Res. Public Health 202017, 5962.

Similarly, the journal has recently published a study demonstrating the importance of strong lower limbs in ultra-runners:

Gajda, R.; Walasek, P.; Jarmuszewski, M. Right Knee—The Weakest Point of the Best Ultramarathon Runners of the World? A Case Study. Int. J. Environ. Res. Public Health 202017, 5955.

Thank you for the amendments included from lines 282 - 293, I appreciate you did not necessarily agree with my comments, but it is pleasing to see them here.

Line 311 - I am happy with the notion that athletes who sustain a higher fractional utilisation of VO2max throughout the race are likely to perform better. This specific wording needs to be included throughout, as this is separate from possessing a high VO2max, in and of itself. I stand by this comment, as in the most comprehensive analysis of VO2max and running economy to date, only small correlations were found between both variables, in both sexes

Shaw, A. J., Ingham, S. A., Atkinson, G., & Folland, J. P. (2015). The correlation between running economy and maximal oxygen uptake: cross-sectional and longitudinal relationships in highly trained distance runners. PloS one10(4), e0123101. https://doi.org/10.1371/journal.pone.0123101

Granted the above is in non-ultra runners, but ultra-runners are more likely to have a metabolism that primes them for economy, as opposed to high VO2max values, given a preponderence of type1 fibres, high rates of fat oxidation etc.

Again, thank you for your revisions, the manuscript is strengthened but I do have some concerns regarding statistical practice, strength of conclusions from the present correlations and sub-group analysis.

Author Response

Thank you for providing a revised version of this paper. I have some amendments that I would prefer you to include before recommending the paper be accepted for publication.

Line 82 - as per my previous review, please consider amending the word better here. My understanding is that you mean faster. It is appropriate to be direct here, especially given that in the next sentence you say that women are better at pacing than male athletes. Thank you for the clarification here though, this is nicely written and highlights the aim of the paper clearly.

Answer: We agree with the expert reviewer and specified the sentence accordingly.

Statistical analyses - I feel that descriptors of correlation coefficients must be included in this section, I wish I had mentioned and pushed for this in my previous review, but was perhaps not firm enough. My rationale for including descriptors in this section is that it makes Table 2 more clear, and highlights that despite the statistical significance of the findings, the relationships between assessed variables are small/weak at best.

Appropriate references for descriptions of correlations can be found:

Cohen, 1988 - 0.1 - 0.3 Small, 0.3 - 0.5 Moderate and >0.5 large

Hinkle et al., 2003 - 0 - 0.3 negligible, 0.3-0.5 Low, 0.5 -0.7 Moderate, 0.7 -0.9 High, >0.9 Very High

Hopkins et al., 2009 - 0 - 0.1 trivial, 0.1 - 0.3 Small, 0.3 - 0.5 Moderate 0.5 - 0.7 Large 0.7 - 0.9 Very Large, >0.9 Nearly Perfect

Answer: Descriptors of correlations were added, as suggested.

My other question regarding this section, is can you analyse top 5 finishers parametrically? By definition, even in a normally distributed data set, the top 5 are extreme outliers, and themselves would not follow a normal distribution, or at least represent an inadequate sample size to make such an assessment. I agree with including them in the paper, and the discussion around these highly placed finishers is worthwhile, but may be statistically flawed.

Answer: We agree with the expert reviewer. But it is noteworthy to mention that when we consider top-5 finishers the total sample for this analysis is has 72 subjects in the TS24, which is lower number of subjects for this analysis. TS18 has n = 228. Thus, we believe that the sample size is adequate for parametric analysis. Moreover, Top-5 as outliers is a matter of perspective. In a general analysis, indeed, top-5 athletes are outliers. However, our analysis is an independent analysis considering ONLY top-5 athletes. This new analysis has its own data distribution.

Thank you for your comments regarding Figure 3D, and further expanding the results section with the inclusion of effect sizes are per partial eta squared. I disagree with the authors' comments that describing the nature of the curves in the figures is of little value, and that is does not agree with the paper's aims. I quote the following from the introduction

'...the present study aimed to analyze how age, sex, and performance level of athletes affect their pacing during the UTMB®, and to identify ideal pacing strategies for ultramarathon distance trail running considering these variables.' 

Describing the curves, as presented by age group (and sex), provides a fuller picture of the data and supports the aim of the paper. If you can see that pace declines more rapidly beyond a certain age, or pace is more variable beyond a certain age, this has the potential to directly influence training and therefore race outcome. Especially, as the authors point out with an increasing participant finishing age, greater participation numbers and the challenging terrain.

Answer: We agree with the expert reviewer and we added further discussion regarding the behavior of each curve.

Discussion - I feel you have mischaracterised the findings and discussion points of reference 36 here. What Millet and colleagues suggest in this paper, is that strategies that may be considered to increase the fractional utilisation of VO2max, at the expense of running economy, may be worth employing IF they reduce the load experienced by the musculotendinous and osteoarticular systems. Again, this is especially important given the aim and novelty of the current paper, as higher loads are placed upon these tissues, with increasing elevation changes throughout a race. Please amend accordingly. 

Answer: We agree with the expert reviewer and changed the discussion accordingly, including the below mentioned publication. As requested, the discussion now includes a more detailed examination of the running economy and VO2max. We are of the opinion, that the changed discussion provides a better explanation of our observations due to the requested papers and we thank the expert reviewer for this.

Indeed further evidence for economy in prolonged running can be found in a paper recently published in this journal:

Berger, N.; Cooley, D.; Graham, M.; Harrison, C.; Best, R. Physiological Responses and Nutritional Intake during a 7-Day Treadmill Running World Record. Int. J. Environ. Res. Public Health 202017, 5962.

Answer: We agree with the expert reviewer and included the mentioned paper in the discussion as stated above.

Similarly, the journal has recently published a study demonstrating the importance of strong lower limbs in ultra-runners:

Gajda, R.; Walasek, P.; Jarmuszewski, M. Right Knee—The Weakest Point of the Best Ultramarathon Runners of the World? A Case Study. Int. J. Environ. Res. Public Health 202017, 5955.

Answer: We agree with the expert reviewer and included the mentioned paper in the discussion as stated above.

Thank you for the amendments included from lines 282 - 293, I appreciate you did not necessarily agree with my comments, but it is pleasing to see them here.

Line 311 - I am happy with the notion that athletes who sustain a higher fractional utilisation of VO2max throughout the race are likely to perform better. This specific wording needs to be included throughout, as this is separate from possessing a high VO2max, in and of itself. I stand by this comment, as in the most comprehensive analysis of VO2max and running economy to date, only small correlations were found between both variables, in both sexes

Shaw, A. J., Ingham, S. A., Atkinson, G., & Folland, J. P. (2015). The correlation between running economy and maximal oxygen uptake: cross-sectional and longitudinal relationships in highly trained distance runners. PloS one10(4), e0123101. https://doi.org/10.1371/journal.pone.0123101

Answer: We agree with the expert reviewer and included the mentioned paper in the discussion as stated above.

Granted the above is in non-ultra runners, but ultra-runners are more likely to have a metabolism that primes them for economy, as opposed to high VO2max values, given a preponderence of type1 fibres, high rates of fat oxidation etc.

Answer: We agree with the expert reviewer and made the suggested amendments. Again, thanks for giving more insight into this topic by the choice of papers.

Again, thank you for your revisions, the manuscript is strengthened but I do have some concerns regarding statistical practice, strength of conclusions from the present correlations and sub-group analysis.